# Integrative Analysis of miR-21, PTEN, and Immune Signatures in Colorectal Cancer

**DOI:** 10.3390/ijms262412118

**Published:** 2025-12-17

**Authors:** Yu-Ting Yen, Chen-I Hsu, Yee-Chun Chen, Shih-Chang Tsai

**Affiliations:** 1Institute of Translational Medicine and New Drug Development, School of Medicine, China Medical University, No. 91, Xueshi Road, Taichung 40402, Taiwan; d92449001@ntu.edu.tw; 2Department of Biological Science and Technology, China Medical University, No. 100, Sec. 1, Jing-Mao Road, Taichung 406040, Taiwan; chenihsu003@gmail.com; 3National Institute of Infectious Diseases and Vaccinology, National Health Research Institutes, R1-7F, No. 35, Keyan Road, Zhunan 35053, Miaoli County, Taiwan; yeechunchen@gmail.com

**Keywords:** colorectal cancer, miR-21-5p, PTEN, PI3K/AKT signaling, immune evasion, immune infiltration

## Abstract

Colorectal cancer (CRC) remains a major cause of cancer-related mortality worldwide. While immune checkpoint blockade (ICB) has transformed cancer therapy, its clinical benefit in CRC is often limited by an immune-excluded tumor microenvironment (TME). MicroRNA-21-5p (miR-21-5p) is a well-established oncomiR in CRC; however, its role in immune resistance remains incompletely elucidated. In this study, we explored the potential immunoregulatory role of miR-21-5p in CRC by integrating transcriptomic profiling of TCGA-COAD and TCGA-READ cohorts with experimental validation of its target PTEN in CRC cell models. MiR-21-5p was markedly upregulated in tumors compared with adjacent normal tissues and was associated with reduced infiltration of CD8^+^ T cells and dendritic cells. Functional assays confirmed that miR-21-5p directly targets PTEN; transcriptomic correlations further suggested potential links to PI3K/AKT activation and alterations in JAK–STAT and Th17-associated signaling. Elevated miR-21-5p was associated with transcriptomic signatures indicative of altered Th1/Th2 balance, reduced IgA-related immune responses, and features of an immune-excluded TME. Therapeutically, the inhibition of miR-21-5p has been reported in previous studies to restore PTEN and modulate signaling pathways. However, our study did not evaluate immune reactivation or checkpoint-blockade efficacy; thus, such therapeutic implications remain hypothetical. Collectively, these findings suggest that the miR-21–PTEN–PI3K/AKT axis may contribute to shaping immune-related features in CRC. These findings provide a rationale for future studies investigating whether targeting miR-21-5p could enhance antitumor immunity or improve immunotherapy response in CRC.

## 1. Introduction

Colorectal cancer (CRC) remains one of the most common and lethal malignancies worldwide, with over 1.9 million new cases and nearly 900,000 deaths annually [1,2]. Although immune checkpoint blockade (ICB) has revolutionized the treatment of CRC [3], this therapeutic resistance is largely attributed to an immunologically “cold” tumor microenvironment (TME) characterized by poor infiltration of cytotoxic CD8^+^ T cells and antigen-presenting cells, ultimately facilitating tumor immune escape [4].

Growing evidence suggests that tumor-intrinsic mechanisms play a central role in shaping the immunosuppressive TME. Among these, microRNAs (miRNAs)—small noncoding RNAs that modulate gene expression post-transcriptionally—have emerged as critical regulators of cancer progression and immune evasion. MicroRNA-21-5p (miR-21-5p) is one of the most consistently overexpressed oncomiRs across multiple tumor types, including CRC [5,6]. It promotes tumor growth, invasion [7], and therapy resistance [8], primarily by repressing tumor suppressors such as the phosphatase and tensin homolog (PTEN) [9].

Beyond its oncogenic roles, miR-21-5p has been implicated in modulating immune responses. Prior studies have shown that miR-21-5p can activate the PI3K/AKT pathway via PTEN downregulation, suppress CD8^+^ T cell activity, and promote regulatory T cell (Treg) expansion and M2 macrophage polarization [10,11]. In CRC tumors, exosomal miR-21-5p has been shown to enhance PD-L1 expression and STAT3 activation [12,13], and to regulate dendritic cell and Treg differentiation via TGF-β-dependent signaling [14,15]. While these findings provide a foundation for understanding the immunoregulatory potential of miR-21, the broader immunologic consequences of this miRNA—particularly in the context of CRC—remain incompletely defined.

In this study, we performed an integrative analysis combining pan-cancer transcriptomic profiling with functional validation in CRC models to elucidate the immunoregulatory role of miR-21-5p. Our analyses indicate that miR-21-5p represses PTEN and activates the PI3K/AKT signaling cascade, leading to downstream disruption of Th17 differentiation, IgA production, and JAK–STAT-mediated immune responses, which are critical components of antitumor immunity. We therefore hypothesize that miR-21-5p establishes a coordinated repression of the PTEN–JAK/STAT–Th17 axis, promoting immune exclusion and resistance to immune checkpoint blockade in CRC. This work aims to provide mechanistic and translational insight into how miR-21-driven PTEN suppression contributes to tumor-intrinsic immune evasion, thereby identifying a potential therapeutic target to restore immune responsiveness in immune-refractory colorectal cancer.

## 2. Results

### 2.1. miR-21-5p Is Highly Expressed in Colorectal Cancer and Exhibits Strong Diagnostic Performance

Analysis of The Cancer Genome Atlas (TCGA) datasets revealed that miR-21-5p ranks among the most abundantly expressed microRNAs in both colon (COAD) and rectal (READ) adenocarcinomas (Appendix A). Tumor tissues displayed markedly elevated miR-21-5p levels compared with matched normal counterparts (Figure 1A,C,D), a pattern that was consistent across several other cancer types, including breast, lung, liver, and head and neck malignancies. Receiver operating characteristic (ROC) analysis demonstrated excellent diagnostic accuracy in terms of differentiating tumors from normal tissues, with an area under the curve (AUC) of 1.00 in CRC cohorts (Figure 1B). These data indicate that miR-21-5p is a robust diagnostic biomarker and a potential molecular signature of colorectal tumorigenesis.

### 2.2. miR-21-5p Is Inversely Associated with Tumor Suppressor Gene Expression and Immune-Related Pathways

Correlation analysis using the LinkedOmics database identified a subset of genes significantly and negatively associated with miR-21-5p expression in CRC (Figure 2). These genes were enriched for regulators of cell adhesion, epithelial differentiation, and immune activation. KEGG enrichment analysis of negatively correlated genes revealed downregulation of immune-related pathways—such as cytokine–cytokine receptor interaction and T cell receptor signaling—alongside the activation of oncogenic cascades, including PI3K/AKT and MAPK signaling (Table 1). Collectively, these data suggest that high miR-21-5p expression may simultaneously suppress antitumor immunity and promote tumor progression.

### 2.3. PTEN Is a Direct Target of miR-21-5p and Is Downregulated in CRC

Computational prediction identified several tumor-suppressor genes, including *PTEN*, *STAT3*, *RhoB*, and *PDCD4*, as potential direct targets of miR-21-5p (Figure 3A). Dual-luciferase reporter assays demonstrated that miR-21-5p significantly reduced luciferase activity from constructs containing the wild-type 3′-untranslated regions (3′-UTRs) of these genes, whereas mutation of the seed sequences abrogated this effect (Figure 3B). Immunoblot analysis confirmed that enforced expression of miR-21-5p in SW620 cells decreased PTEN protein levels, along with the downregulation of additional predicted targets (Figure 3C), supporting a post-transcriptional suppressive mechanism.

### 2.4. High miR-21-5p and Low PTEN Expression Are Associated with Impaired Immune Activation and Unfavorable Prognosis

Integrative transcriptomic analyses revealed an inverse correlation between miR-21-5p and *PTEN* expression in CRC specimens (Figure 4A,C). Tumors with high miR-21-5p and concomitantly low PTEN levels exhibited reduced activity in terms of Th17 differentiation, IgA production, and JAK–STAT signaling pathways (Figure 4B,D). This molecular signature was particularly pronounced in the READ cohort; however, the association with overall survival did not reach statistical significance (Figure 5A,B). These observations suggest that miR-21-5p-mediated PTEN suppression may be linked to immune-related transcriptional patterns; however, its relationship to clinical outcomes remains inconclusive in our dataset.

### 2.5. PTEN Expression Positively Correlates with Immune Cell Infiltration

Evaluation using the TIMER platform demonstrated that PTEN expression is positively correlated with the abundance of CD8^+^ T cells, dendritic cells, and macrophages in CRC (Figure 6A and Table 2). In contrast, tumors characterized by PTEN downregulation—frequently associated with elevated miR-21-5p—showed reduced immune cell infiltration (Figure 6B). These results reflect transcriptomic associations and should not be interpreted as evidence that PTEN directly regulates immune infiltration. Instead, the data indicate that PTEN expression coincides with immune-related patterns in CRC, and further functional studies will be required to determine whether PTEN influences tumor–immune interactions.

### 2.6. Proposed Model of miR-21-5p-Mediated Immune Evasion in CRC

Based on integrative bioinformatics and experimental evidence, we propose a mechanistic model in which miR-21-5p overexpression suppresses PTEN, leading to the activation of the PI3K/AKT signaling cascade and the downstream inhibition of JAK-STAT-dependent immune pathways (Figure 7). This signaling rewiring was associated with transcriptomic signatures related to Th17 differentiation and IgA pathways; these observations reflect correlation and do not establish reduced IgA production or altered mucosal immunity. Collectively, the miR-21–PTEN–PI3K/AKT axis orchestrates both oncogenic signaling and immune evasion, establishing miR-21-5p as a dual driver of tumor progression and immunotherapy resistance in CRC.

## 3. Discussion

### 3.1. Tumor-Intrinsic Immune Suppression Mediated by miR-21-5p in Colorectal Cancer

This study provides mechanistic evidence that miR-21-5p functions as a molecular hub integrating oncogenic signaling and immune suppression in colorectal cancer (CRC). Consistent with prior observations, miR-21-5p is markedly overexpressed in CRC [16] and in multiple other malignancies, including those of the breast [17], lung [18], and liver [19], underscoring its role as a prototypical oncomiR.

miR-21-5p directly represses the tumor suppressor *PTEN*, leading to sustained activation of the PI3K/AKT pathway. This cascade promotes tumor cell survival and proliferation. Transcriptomic correlations further suggest associations with variation in Th1/Th2- [20], Th17- [16], and JAK–STAT-related signatures [21]; however, these immunologic effects were not experimentally validated in this study. Consistent with previous findings that miR-21 is transcriptionally induced by STAT3 and interferon signaling [22], and that it regulates immune activation through the PDCD4-dependent modulation of cytokine responses [23], our data indicate that miR-21-5p reinforces this feedback loop, thereby amplifying immune-suppressive signaling. Collectively, these alterations were associated with transcriptomic features often seen in immune-excluded CRC; however, this study did not perform functional or spatial analyses to confirm an immune-excluded phenotype.

In addition to PTEN, miR-21-5p targets other tumor suppressors such as PDCD4 [24] and SPRY2 [25], further potentiating oncogenic signaling and metastatic potential [26]. Notably, our results suggest that the miR-21/*PTEN* axis may constitute a conserved immunosuppressive circuit operating across cancer and infection. Supporting this, evidence from *Candida albicans* infection models demonstrates that monocyte-derived miR-21-5p diminishes mucosal IgA secretion while promoting Treg and M2-macrophage polarization [27]. The reduced intestinal IgA expression observed in our CRC datasets (Appendix A) supports this shared mechanism, highlighting the pleiotropic immunomodulatory roles of miR-21-5p.

Moreover, the observed reduction in Th17- and IgA-related gene expression signatures in bulk RNA-seq likely reflects impaired mucosal immune surveillance. Validation in organoid-based or co-culture systems will be instrumental in delineating the cellular specificity of these effects. Finally, miR-21-5p expression appears to be dynamically regulated by proinflammatory cytokines such as IL-6 and TGF-β, suggesting that inflammatory cues within the tumor microenvironment further enhance its induction. Such cytokine-driven induction may represent an exploitable temporal window for therapeutic intervention, particularly in targeting cancer stem cell-associated immune resistance, where checkpoint blockade alone remains largely ineffective.

### 3.2. Exosomal miR-21-5p and Systemic Immune Modulation

In addition to its intracellular functions, miR-21-5p is actively secreted through tumor-derived exosomes and exerts paracrine and systemic immunomodulatory effects. Previous studies have shown that exosomal miR-21-5p polarizes macrophages toward an M2 phenotype [27,28] and promotes Treg expansion, thereby contributing to systemic immunosuppression. Our integrative analyses further link the miR-21/PTEN/PI3K–AKTaxis to the inhibition of IFN-γ responses, antigen-processing pathways, and cytotoxic T cell activation [29]. These findings align with recent reports highlighting exosomal miRNAs as critical mediators of tumor–immune crosstalk [30].

Moreover, exosomal miR-21-5p impairs dendritic cell maturation and facilitates TGF-β-dependent differentiation of Tregs, reinforcing immune tolerance within the TME. These findings position miR-21-5p as both an intracellular oncogenic driver and an extracellular “immune rheostat” that fine-tunes host antitumor responses.

### 3.3. Therapeutic Implications for CRC

The extracellular stability of miR-21-5p, whether via exosomal encapsulation or Argonaute association, supports its role as a liquid biopsy biomarker. Therapeutically, inhibition of miR-21-5p restores PTEN, reactivates immune signaling, and enhances the efficacy of checkpoint blockade. Preclinical studies employing locked nucleic acid (LNA) antagomirs [31] or small-molecule inhibitors [32] have successfully reinstated *PTEN* expression and attenuated PI3K/AKT signaling.

Combination strategies targeting miR-21-5p alongside PD-1/PD-L1 blockade have demonstrated synergistic antitumor effects in preclinical models [33], supporting the concept that reversing miR-21-driven immune suppression can potentiate checkpoint inhibitor efficacy. Integrating miR-21-targeted therapy with ICB could counteract CSC-associated immune resistance and promote a more immune-inflamed tumor phenotype, warranting further preclinical testing in syngeneic CRC models [34]. Future studies should explore LNA-anti-miR-21 delivery combined with PD-1 blockade in syngeneic CRC models to validate the effects of immune reprogramming.

### 3.4. Future Perspectives and Limitations

While this study delineates the miR-21–PTEN–PI3K/AKT immune evasion axis, further validation in immunocompetent models remains warranted. Although our gain-of-function analyses established that miR-21-5p represses PTEN, complementary loss-of-function and PTEN rescue experiments are required to provide definitive causal evidence. Future work should also incorporate single-cell and spatial transcriptomic profiling to elucidate compartment-specific regulation of the miR-21/PTEN pathway within the tumor microenvironment. In addition, systematic assessment of off-target effects, delivery efficiency, and pharmacokinetics will be essential to the translational development of miR-21-targeted therapeutics. Collectively, these investigations will further substantiate the therapeutic potential of targeting miR-21-5p to restore immune responsiveness in CRC.

## 4. Materials and Methods

### 4.1. Data Collection and Bioinformatics Analysis

Public data acquisition

High-throughput RNA sequencing data, miRNA expression profiles, and the corresponding clinical annotations for colorectal cancer (CRC) were obtained from The Cancer Genome Atlas (TCGA; https://www.cancer.gov/tcga, accessed on 6 December 2025) and the Gene Expression Omnibus (GEO; https://www.ncbi.nlm.nih.gov/geo/, accessed on 6 December 2025). The datasets included paired tumor and adjacent normal tissues with matched miRNA and mRNA expression matrices as well as clinicopathological metadata.

### 4.2. miRNA and mRNA Expression Analysis

Differential expression of key miRNAs and their predicted target genes was validated using the CancerMIRNome database (http://bioinfo.jialab-ucr.org/CancerMIRNome, accessed on 6 December 2025), which integrates miRNA expression across TCGA cohorts. Corresponding mRNA levels were assessed via Gene Expression Profiling Interactive Analysis 2 (GEPIA2; http://gepia2.cancer-pku.cn, accessed on 6 December 2025), based on TCGA and GTEx reference datasets.

### 4.3. LinkedOmics Analysis

The LinkedOmics platform (https://www.linkedomics.org, accessed on 6 December 2025) was used to identify genes correlated with miR-21-5p expression in the TCGA-COAD and TCGA-READ datasets. The LinkFinder module calculated Pearson’s correlation coefficients, whereas the LinkInterpreter module performed Gene Set Enrichment Analysis (GSEA) and Kyoto Encyclopedia of Genes and Genomes (KEGG) pathway analyses. Pathways with *p* < 0.05 were considered significantly enriched. These analyses revealed biological functions and signaling cascades associated with miR-21-5p expression in CRC [35].

### 4.4. UALCAN Expression and Prognostic Analysis

UALCAN (http://ualcan.path.uab.edu, accessed on 6 December 2025) was used to analyze level-3 TCGA RNA-seq data to compare miR-21-5p expression in CRC and normal tissues. Expression was further stratified by clinical variables such as cancer stage and nodal status. Kaplan–Meier survival analyses were conducted within UALCAN to assess the prognostic relevance of miR-21-5p expression [35].

### 4.5. Immune Infiltration Analysis

The TIMER web resource (https://cistrome.shinyapps.io/timer, accessed on 6 December 2025) was employed to examine the relationship between *PTEN* expression and immune cell infiltration in CRC. Correlations were evaluated for six major immune cell subsets—CD8^+^ T cells, CD4^+^ T cells, B cells, macrophages, neutrophils, and dendritic cells—using TCGA-derived data. The correlation strength and statistical significance were calculated and visualized as scatter plots [36].

### 4.6. Survival Analysis

Overall survival (OS) analyses were conducted using the Kaplan–Meier Plotter (http://kmplot.com, accessed on 6 December 2025) for TCGA-COAD and TCGA-READ cohorts. Survival curves were stratified by miR-21-5p or *PTEN* expression levels. Hazard ratios (HRs) with 95% confidence intervals (CIs) were derived using the Cox proportional hazards model, adjusting for age, sex, and tumor stage.

### 4.7. Cell Culture and Transfection

The human colorectal cancer cell line SW620 (ATCC, Manassas, VA, USA) was cultured in Dulbecco’s modified Eagle’s medium (DMEM; Gibco, Waltham, MA, USA) supplemented with 10% heat-inactivated fetal bovine serum (FBS), 100 U/mL penicillin G, 100 µg/mL streptomycin, and 0.25 µg/mL amphotericin B. The cells were maintained at 37 °C in a humidified 5% CO_2_ incubator. For transient transfection, miR-21-5p mimics, inhibitors, or negative controls (GenePharma, Shanghai, China) were introduced using Lipofectamine™ 3000 (Invitrogen, Carlsbad, CA, USA) according to the manufacturer’s instructions.

### 4.8. Dual-Luciferase Reporter Assay

To confirm direct binding between miR-21-5p and its predicted target genes, a dual-luciferase reporter assay was performed. The 3′-untranslated regions (3′-UTRs) of *PTEN*, *STAT3*, *RhoB*, *PDCD4*, and *BCL2* containing the putative miR-21-5p binding sites were cloned downstream of the luciferase gene in the pMIR-REPORT vector (Applied Biosystems, Foster City, CA, USA), as previously described [37]. SW620 cells were co-transfected with luciferase constructs, miR-21-5p mimic or control mimic, and a Renilla luciferase plasmid (internal control). After 48 h, luciferase activity was measured using the Dual-Luciferase Reporter Assay System (Promega, Madison, WI, USA). Firefly luciferase signals were normalized to Renilla luciferase to control for transfection efficiency. A significant reduction in normalized activity indicated direct targeting of the respective 3′-UTR.

### 4.9. Western Blot

Protein expression of *PTEN* and downstream effectors was examined by immunoblotting. SW620 cells transfected with miR-21-5p mimics or inhibitors were lysed in RIPA buffer containing protease and phosphatase inhibitors. Protein concentrations were quantified using the Bradford assay. Equal amounts of protein (30–50 µg) were resolved on 10% SDS-PAGE gels and transferred onto PVDF membranes (Millipore, Burlington, MA, USA). Membranes were blocked with 5% non-fat milk in TBST for 1 h and incubated overnight at 4 °C with primary antibodies against PTEN, STAT3, PDCD4, RhoB, BCL-2 (GeneTex, Irvine, CA, USA), and GAPDH (Sigma-Aldrich, St. Louis, MO, USA). Following incubation with HRP-conjugated secondary antibodies, protein bands were visualized using an enhanced chemiluminescence (ECL) detection system (GE Healthcare, Chicago, IL, USA). Band intensities were quantified in ImageJ (version 1.54), and relative protein expression levels were normalized to GAPDH and presented as fold changes relative to control samples.

### 4.10. Statistical Analysis

All experiments were performed in triplicate unless otherwise specified. Data are presented as mean ± standard error of the mean (SEM). Comparisons between two groups were made using unpaired two-tailed Student’s *t*-tests, and multiple-group comparisons employed one-way ANOVA followed by Tukey’s post hoc test where appropriate. Statistical analyses and graphical visualizations were performed using GraphPad Prism 7.0 (GraphPad Software, La Jolla, CA, USA). False discovery rate (FDR) correction was applied to pathway enrichment analyses using the Benjamini–Hochberg method (FDR < 0.05). *p*-values < 0.05 were considered statistically significant.

## 5. Conclusions

In summary, these findings suggest potential associations between the miR-21–PTEN–PI3K/AKT axis and immune-related features in CRC; no significant prognostic value of PTEN expression was demonstrated in this cohort. Through integrative transcriptomic analyses and validation of PTEN as a direct target of miR-21-5p, our findings point to potential associations between miR-21-5p dysregulation, PTEN suppression, and immune-related pathways such as Th17 differentiation, IgA signatures, and JAK–STAT-associated signaling. These transcriptomic associations may suggest immune-related patterns in CRC; however, no causal mechanisms or effects on immune-checkpoint response were demonstrated in this study.

Our findings suggest that miR-21-5p may serve as a biomarker associated with immune-related signatures in CRC, but its functional role in immune escape remains to be determined through additional mechanistic studies. Therapeutically, targeting miR-21-5p represents a potential approach that warrants future investigation to determine whether restoring PTEN function could modulate antitumor immunity or affect responses to immunotherapy in CRC. Overall, these results provide hypothesis-generating insights that may guide future studies evaluating whether modulation of the miR-21–PTEN–PI3K/AKT axis can influence the tumor immune microenvironment or enhance immunotherapeutic responses in colorectal cancer.

## Figures and Tables

**Figure 1 ijms-26-12118-f001:**
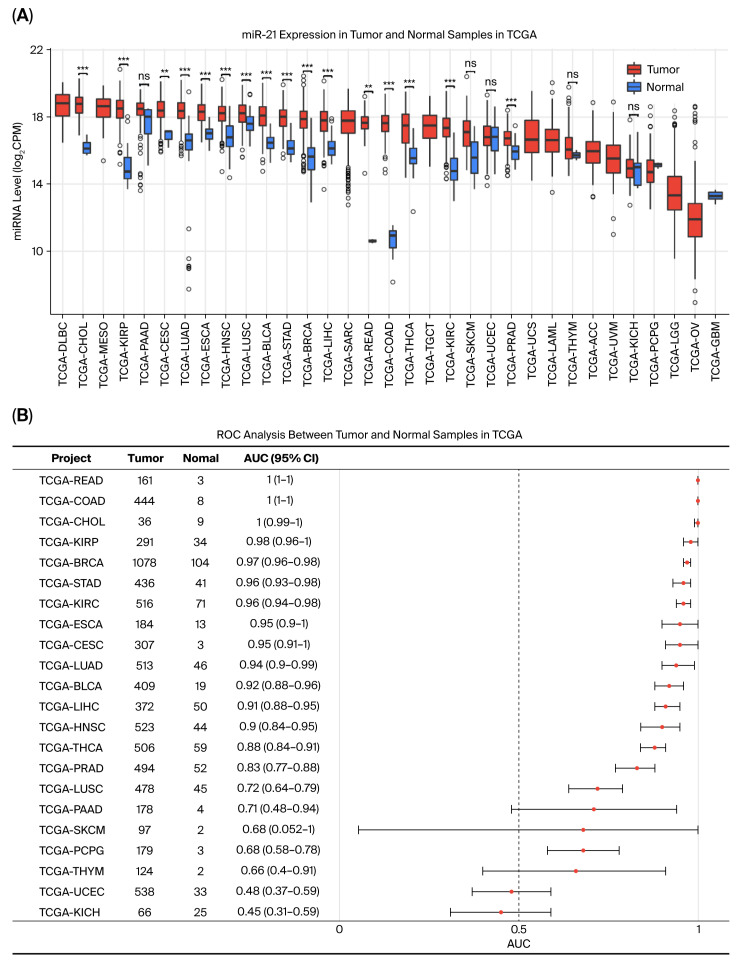
Pan-cancer and CRC-specific expression of miR-21-5p. (**A**) Pan-cancer analysis of miR-21-5p expression (Y axis, log_2_ counts per million, CPM) across TCGA tumor types reveals upregulation of miR-21-5p in the majority of malignancies evaluated, including colon adenocarcinoma and rectal adenocarcinoma (TCGA-COAD and TCGA-READ on X-axis, respectively). Statistical comparisons between tumor and normal tissues were performed using the Wilcoxon rank sum test. Statistical significance is denoted as follows: ns, non-significantly difference; **, *p* < 0.01, and ***, *p* < 0.001. (**B**) High diagnostic accuracy of miR-21-5p in distinguishing CRC tumors from adjacent normal tissues based on receiver operating characteristic (ROC) curve analysis. Numbers of tumors and normal tissues analyzed in TCGA datasets are provided. AUC, area under the curve; 95% CI, 95% confidence interval. The AUC quantifies the overall performance of the test, with values closer to 1 indicating better discrimination. (**C**,**D**) miR-21-5p expression is significantly elevated in tumor tissues compared to matched normal tissues in both colon adenocarcinoma ((**C**) panel, COAD) and rectal adenocarcinoma ((**D**) panel, READ) cohorts. Bar plots displaying mean expression (Y axis, log_2_ counts per million, CPM) of miR-21-5p.

**Figure 2 ijms-26-12118-f002:**
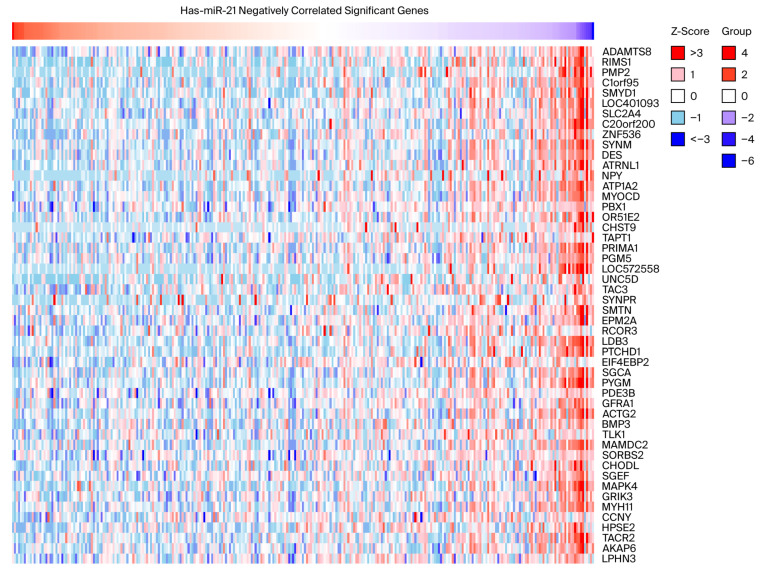
Transcriptomic networks associated with miR-21-5p in colorectal cancer. Heatmap showing the top 50 genes negatively correlated with miR-21-5p expression in the TCGA-COADREAD cohort, as identified using LinkedOmics.

**Figure 3 ijms-26-12118-f003:**
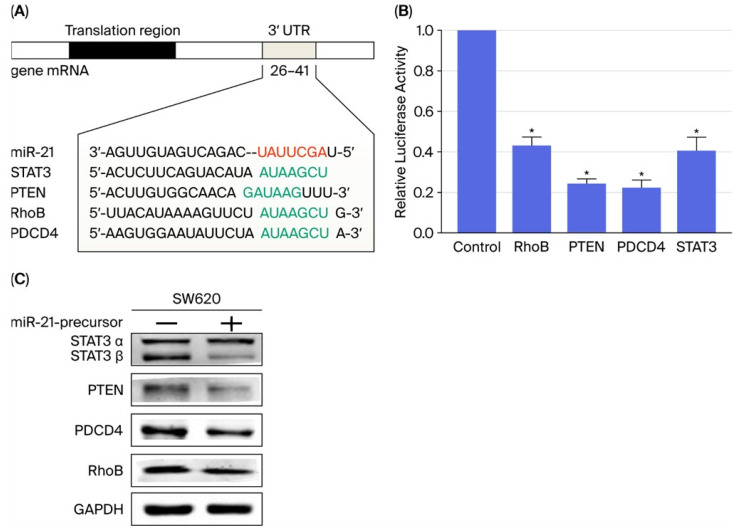
Direct targeting of tumor suppressors by miR-21-5p. (**A**) Bioinformatic prediction of miR-21-5p binding sites in the 3′-UTRs of tumor suppressor genes using TargetScan. The colored nucleotides indicate that green represents the seed sequence and red represents the target regions. (**B**) Dual-luciferase reporter assays in SW620 cells show reduced luciferase activity upon co-transfection with miR-21-5p and 3′-UTR constructs of PTEN, STAT3, RhoB, and PDCD4. *, *p* < 0.05. (**C**) Western blot analysis confirms reduced protein expression of the same targets following miR-21-5p overexpression.

**Figure 4 ijms-26-12118-f004:**
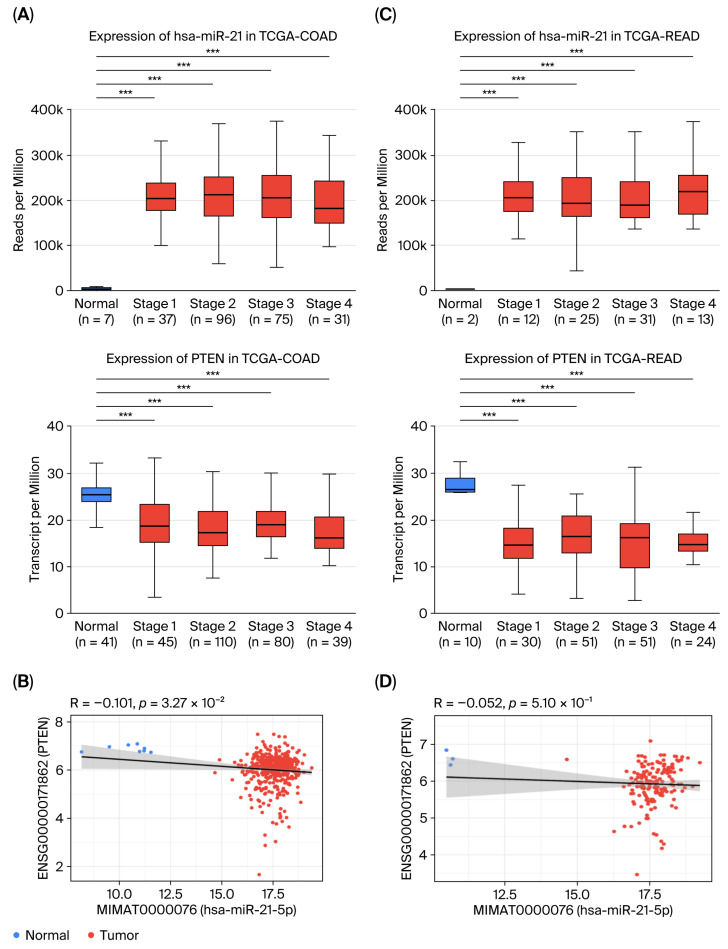
Inverse correlation between miR-21-5p and PTEN in CRC. (**A**,**C**) Expression levels of miR-21-5p and PTEN stratified by tumor stage in the TCGA-COAD and READ cohorts. ***, *p* < 0.001. (**B**,**D**) Scatter plots reveal a significant negative correlation between miR-21-5p and PTEN expression in both COAD and READ. The blue and red dots represent normal and tumor samples, respectively.

**Figure 5 ijms-26-12118-f005:**
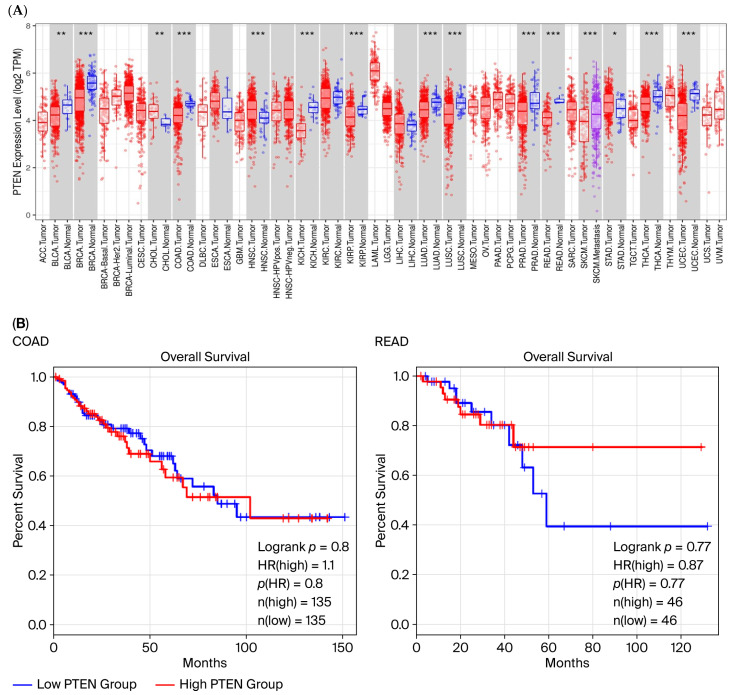
Clinical significance of PTEN in colorectal cancer. (**A**) Differential expression analysis shows decreased PTEN transcript levels in CRC tumor tissues compared to normal tissues. *, *p* < 0.05, **, *p* < 0.01, ***, *p* < 0.001. (**B**) Kaplan–Meier survival curves illustrate overall survival stratified by PTEN expression levels; no statistically significant differences were observed between groups. Statistical significance was assessed using Welch’s *t*-test and the log-rank test (*p* = 4 × 10^−4^). Sample sizes are indicated in the figure panels.

**Figure 6 ijms-26-12118-f006:**
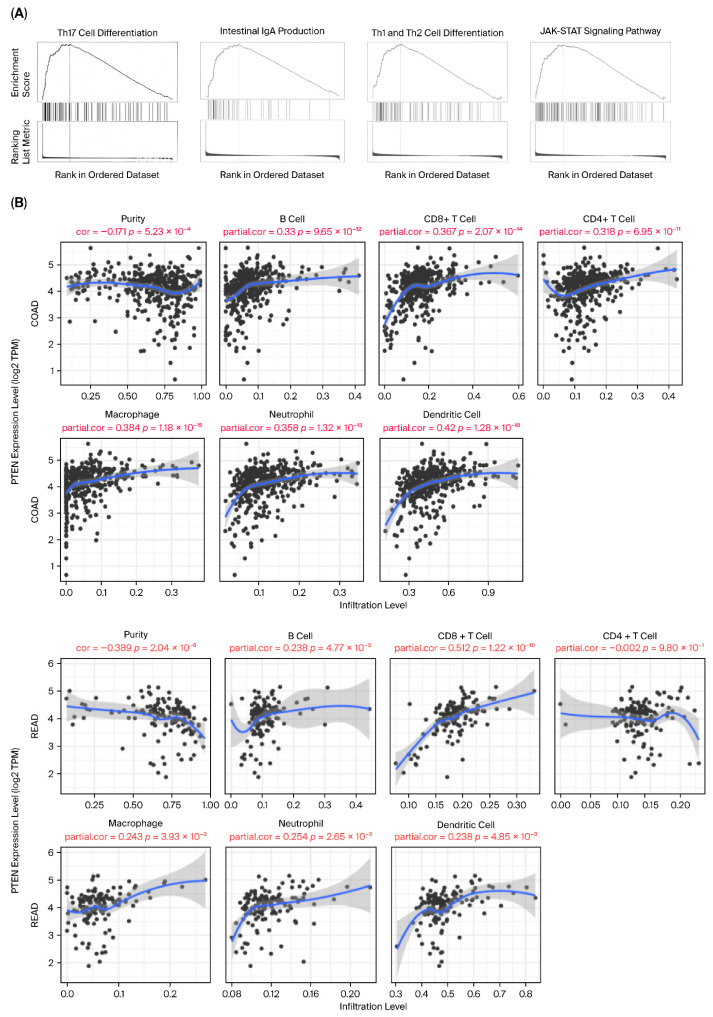
PTEN expression correlates with immune signaling and cell infiltration. (**A**) GSEA reveals downregulation of key immune pathways, including Th17 cell differentiation, IgA production, Th1/Th2 differentiation, and JAK–STAT signaling, in PTEN-low tumors. (**B**) Correlation analysis demonstrates that PTEN expression is positively associated with infiltration by B cells, CD8^+^ T cells, CD4^+^ T cells, neutrophils, macrophages, and dendritic cells in TCGA-COAD and READ datasets.

**Figure 7 ijms-26-12118-f007:**
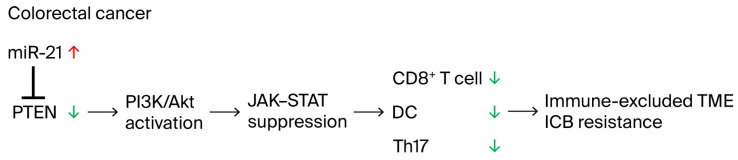
Proposed model of miR-21-5p-mediated immune evasion. Schematic illustration of the molecular mechanism by which miR-21-5p binds the 3′-UTR of PTEN mRNA, resulting in translational repression. The colored arrows indicate expression changes: red arrows represent upregulation, and green arrows represent downregulation. Downstream effects include activation of the PI3K/AKTpathway, reduced immune cell infiltration, altered Th17 cell development, and suppression of JAK–STAT signaling, collectively contributing to immune escape and tumor progression in CRC. Feedback activation of STAT3 by tumor-derived cytokines may further enhance miR-21 expression, forming a self-reinforcing immunosuppressive loop.

**Table 1 ijms-26-12118-t001:** Top KEGG pathways enriched in the miR-21 gene signature identified by GSEA.

Description	Size	Leading Edge Number	ES	NES	*p* Value	FDR
Antigen processing and presentation	68	40	0.74072	2.4975	<2.2 × 10^−16^	<2.2 × 10^−16^
Hematopoietic cell lineage	93	49	0.67393	2.4705	<2.2 × 10^−16^	<2.2 × 10^−16^
Toll-like receptor signaling pathway	96	38	0.66632	2.4394	<2.2 × 10^−16^	<2.2 × 10^−16^
Osteoclast differentiation	126	47	0.63476	2.4287	<2.2 × 10^−16^	<2.2 × 10^−16^
Phagosome	144	51	0.6074	2.3955	<2.2 × 10^−16^	<2.2 × 10^−16^
NOD-like receptor signaling pathway	157	61	0.60571	2.3663	<2.2 × 10^−16^	<2.2 × 10^−16^
Cytosolic DNA-sensing pathway	55	26	0.70094	2.2973	<2.2 × 10^−16^	<2.2 × 10^−16^
Natural killer cell-mediated cytotoxicity	117	52	0.61234	2.2788	<2.2 × 10^−16^	<2.2 × 10^−16^
Proteasome	44	33	0.70113	2.2401	<2.2 × 10^−16^	<2.2 × 10^−16^
Cytokine–cytokine receptor interaction	274	96	0.52239	2.1771	<2.2 × 10^−16^	<2.2 × 10^−16^
Insulin secretion	85	26	−0.60735	−1.9731	<2.2 × 10^−16^	<2.2 × 10^−16^
Circadian entrainment	94	31	−0.60202	−1.9738	<2.2 × 10^−16^	<2.2 × 10^−16^
cAMP signaling pathway	193	46	−0.52073	−1.8511	<2.2 × 10^−16^	0.0024103
cGMP-PKG signaling pathway	160	59	−0.51717	−1.7921	<2.2 × 10^−16^	0.0039441

**Table 2 ijms-26-12118-t002:** Immune-related KEGG pathways associated with the miR-21 gene signature identified by GSEA.

Description	Size	Leading Edge Number	ES	NES	FDR
Th17 cell differentiation	105	55	0.61623	2.0455	0.005937
Intestinal immune network for IgA production	45	32	0.68539	2.0104	0.007037
Th1 and Th2 cell differentiation	90	47	0.61978	1.984	0.007751
JAK-STAT signaling pathway	152	58	0.56912	1.9477	0.010027
Cytokine–cytokine receptor interaction	275	114	0.53498	1.9211	0.010005

## Data Availability

The original contributions presented in this study are included in the article/Appendix A. Further inquiries can be directed to the corresponding author. All data analyzed in this study were obtained from publicly accessible repositories, including The Cancer Genome Atlas (TCGA), the Gene Expression Omnibus (GEO), and the CancerMIRNome database. Accession identifiers and dataset details are provided in Section 4.

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
