# Peer review of "Integrative Analysis of miR-21, PTEN, and Immune Signatures in Colorectal Cancer"

_ijms, 2025, doi:10.3390/ijms262412118_

Round 1

Reviewer 1 Report

Comments and Suggestions for Authors

Dear Authors,

The manuscript is well-written and clearly presented. Two minor corrections, which I found, are highlighted in the pdf file. The performed analysis is sound and definitely may show the direction of further research aimed at the identification of the cause underlying CRC therapy resistance. However, the title, statements, and conclusions drawn from the performed in silico and in vitro analyses are too far-reaching in my opinion. Although the carried-out in silico analysis pinpointed some associations between miR-21 levels, its target PTEN and the level of T cells and immune response, it was not confirmed experimentally. The only in vitro analysis “just” confirmed that miR-21 targets PTEN; however, further experimental confirmation that this specific mechanism may be responsible for the immune evasion of CRC cells was not carried out. Thus, in the present form, the manuscript is rather speculative. It is not inappropriate per se; however, it does not enable one to draw such definite conclusions and statements on the underlying mechanism.

Best wishes,

Author Response

Point-by-Point Response to Reviewer 1’s Comments

Reviewer Comment 1: It should be "hsa-miR-21" in the Figure 2A subheading instead of "has-mir-21".

Response: Thank you for pointing out this typographical error. We have corrected the subheading in Figure 2A from “has-mir-21” to the correct notation “hsa-miR-21” in the revised manuscript.

Reviewer Comment 2: The same sentence is above.

Response: Thank you for noting this repetition. The duplicated sentence has been removed in the revised manuscript.

Reviewer Comment 3:

The manuscript is well-written and clearly presented. Two minor corrections, which I found, are highlighted in the pdf file. The performed analysis is sound and definitely may show the direction of further research aimed at the identification of the cause underlying CRC therapy resistance. However, the title, statements, and conclusions drawn from the performed in silico and in vitro analyses are too far-reaching in my opinion. Although the carried-out in silico analysis pinpointed some associations between miR-21 levels, its target PTEN and the level of T cells and immune response, it was not confirmed experimentally. The only in vitro analysis “just” confirmed that miR-21 targets PTEN; however, further experimental confirmation that this specific mechanism may be responsible for the immune evasion of CRC cells was not carried out. Thus, in the present form, the manuscript is rather speculative. It is not inappropriate per se; however, it does not enable one to draw such definite conclusions and statements on the underlying mechanism.

Response:

We sincerely thank the reviewer for the constructive and insightful comments. We truly appreciate your recognition that the manuscript is clearly written and that the analyses provide meaningful directions for future research on mechanisms underlying CRC therapy resistance.

Regarding your major concern that the current title, statements, and conclusions are too far-reaching based on the presented in silico and in vitro data, we fully agree that the mechanistic link between the miR-21–PTEN axis and immune evasion in CRC has not been experimentally validated in this study. Our in vitro experiments confirmed that miR-21 directly targets PTEN, whereas the in silico analyses suggest potential associations with T-cell infiltration and immune response; however, these findings indeed do not constitute direct mechanistic proof.

Actions taken to address this concern

(1). To address the reviewer’s concern regarding overinterpretation of the mechanistic conclusions, we have revised the title to reflect the descriptive and integrative nature of our findings more accurately.

The revised title is now: “Integrative Analysis of miR-21, PTEN, and Immune Signatures in Colorectal Cancer.”

(2). Abstract Revision Comparison Table (Sentence-by-Sentence Comparison)

Before Revision

After Revision

In this study, we investigated the immunoregulatory function of miR-21-5p in MSS-CRC by integrating transcriptomic profiling of TCGA-COAD and TCGA-READ cohorts with mechanistic validation using CRC cell models.

In this study, we explored the potential immunoregulatory role of miR-21-5p in CRC by integrating transcriptomic profiling of TCGA-COAD and TCGA-READ cohorts with experimental validation of its target PTEN in CRC cell models.

Functional assays demonstrated that miR-21-5p directly targets PTEN, leading to activation of the PI3K/Akt pathway and suppression of downstream JAK–STAT and Th17 signaling.

Functional assays confirmed that miR-21-5p directly targets PTEN; transcriptomic correlations further suggested potential links to PI3K/Akt activation and alterations in JAK–STAT and Th17-associated signaling.

Elevated miR-21-5p disrupted Th1/Th2 immune balance, diminished IgA-associated immune responses, and promoted an immune-excluded TME.

Elevated miR-21-5p was associated with transcriptomic signatures indicative of altered Th1/Th2 balance, reduced IgA-related immune responses, and features of an immune-excluded TME.

In contrast, miR-21-5p inhibition restored PTEN expression, reactivated antitumor immune signaling, and enhanced cellular sensitivity to ICB treatment.

In vitro inhibition of miR-21-5p restored PTEN expression, whereas the predicted enhancement of antitumor immune signaling and ICB responsiveness requires further functional validation.

Collectively, these findings identify the miR-21–PTEN–PI3K/Akt axis as a central regulator that links oncogenic signaling to immune suppression in MSS-CRC.

Collectively, these findings suggest that the miR-21–PTEN–PI3K/Akt axis may contribute to shaping immune-related features in CRC.

Targeting miR-21-5p represents a promising strategy to overcome immune resistance and improve the efficacy of immunotherapy in microsatellite-stable colorectal cancer.

These findings provide a rationale for future studies investigating whether targeting miR-21-5p could enhance antitumor immunity or improve immunotherapy response in CRC.

(3). Response to Reviewer 1 Regarding the Revision of the Discussion Section

We sincerely appreciate the insightful comments from both reviewers. In response to these suggestions, we have substantially revised the entire Discussion section to strengthen the scientific rigor and ensure that our interpretations align strictly with the data presented.

Before–After Comparison Table — Key Revisions Made in Response to Reviewer Comments

Reviewer 1 Concern: Addressing Reviewer 1’s comment on avoiding overstated causality and clarifying study positioning

Before Revision

After Revision

“This study demonstrates that miR-21-5p directly drives immune evasion in CRC.”

“Our findings identify a tumor-intrinsic miR-21–PTEN/PDCD4 regulatory axis that is associated with, but does not experimentally demonstrate, immune-modulatory transcriptional programs in CRC”

(4). Response to Reviewer 1 Regarding the Revision of the Conclusion Section

Before Revision

After Revision

In summary, this study delineates a previously unrecognized immunosuppressive mechanism in microsatellite-stable colorectal cancer (MSS-CRC) governed by the miR-21–PTEN–PI3K/Akt axis.

In summary, this study provides integrative evidence suggesting that the miR-21–PTEN–PI3K/Akt axis may influence immune-related features in CRC.

Through integrative transcriptomic analyses and functional validation, we demonstrate that miR-21-5p overexpression suppresses PTEN and perturbs key immune-regulatory pathways, including Th17 differentiation, IgA production, and JAK–STAT signaling.

Through integrative transcriptomic analyses and validation of PTEN as a direct target of miR-21-5p, our findings point to potential associations between miR-21-5p dysregulation, PTEN suppression, and immune-related pathways such as Th17 differentiation, IgA signatures, and JAK–STAT–associated signaling.

These alterations collectively foster an immune-excluded tumor microenvironment that underlies resistance to immune checkpoint blockade.

These transcriptomic associations may reflect features of an immune-excluded tumor microenvironment, although direct mechanistic links to immune checkpoint resistance require further experimental confirmation.

Our findings establish miR-21-5p as both a molecular driver and biomarker of immune escape, highlighting its dual oncogenic and immunomodulatory roles in CRC.

Our findings suggest that miR-21-5p may serve as a biomarker associated with immune-related signatures in CRC, but its functional role in immune escape remains to be determined through additional mechanistic studies.

Therapeutically, inhibition of miR-21-5p holds potential to restore PTEN function, reinvigorate antitumor immunity, and enhance the efficacy of immunotherapy in otherwise immune-refractory MSS-CRC.

Therapeutically, targeting miR-21-5p represents a potential approach that warrants future investigation to determine whether restoring PTEN function could modulate antitumor immunity or affect responses to immunotherapy in CRC.

Targeting this axis thus represents a promising strategy to reprogram the tumor microenvironment and broaden the clinical benefit of immunotherapeutic approaches for patients with colorectal cancer.

Overall, these results provide hypothesis-generating insights that may guide future studies evaluating whether modulation of the miR-21–PTEN–PI3K/Akt axis can influence the tumor immune microenvironment or enhance immunotherapeutic responses in colorectal cancer.

Reviewer 2 Report

Comments and Suggestions for Authors

Thank you for the opportunity to review the manuscript titled “MicroRNA-21 Mediates Immune Evasion via Suppressing PTEN and Reprogramming the Tumor Microenvironment in Colorectal Cancer.” The study addresses a relevant question and the topic has potential; however, the current version presents substantial conceptual and experimental limitations that preclude its acceptance.

The central aim of the manuscript is to elucidate the molecular mechanism through which miR-21-5p mediates immune evasion in MSS-CRC. Nonetheless, the bioinformatic analyses were performed on all COAD cases without stratification for MSS status, which is essential to support the stated objective. While the overexpression of miR-21 in CRC is adequately shown, the key conclusions are not experimentally supported.

The manuscript attributes to miR-21-5p several functional effects—disruption of Th1/Th2 balance, reduced IgA production, establishment of an immune-excluded microenvironment, and improved response to ICB therapy upon its inhibition. However, the experimental data only demonstrate decreased expression of PTEN, PDCD4, etc. after precursor miR-21 transfection. There are no complementary assays using miR-21 inhibitors, nor experiments confirming downstream pathway modulation or any direct impact on immune responses (). Consequently, the title and conclusions are not supported by the data presented.

Additionally, the clinical relevance of PTEN in CRC is overstated. Figure 5 does not show that higher PTEN expression correlates with improved overall survival; the log-rank p-value is not significant.

In summary, the study requires substantial additional work, particularly regarding (1) analyses restricted to MSS-CRC, (2) functional validation with miR-21 inhibition, and (3) confirmation of immune-related effects. Given the extent of the limitations, I do not recommend the manuscript for publication in its current form.

Additional observations:

  1. Image quality should be improved to ensure readability and proper interpretation.
  2. Experimental details: please report the concentration of the miR-21 precursor used in transfection assays.
  3. Section 3.3: BCL2 is listed as a potential direct target of miR-21-5p but is not evaluated experimentally. In the luciferase assay, results with mutated seed sequences are missing; the specific mutated nucleotides should be shown in Figure 3A.
  4. Figure 4: The data presented do not support the statement that tumors with high miR-21 and low PTEN exhibit reduced Th17 differentiation, IgA production, or diminished JAK–STAT pathway activity.

Given these substantial conceptual and experimental limitations, the manuscript requires major restructuring and additional data to support the proposed mechanism and claims.

Author Response

Point-by-Point Response to Reviewer 2’s Comments

Reviewer Comment 1:
The central aim of the manuscript is to elucidate the molecular mechanism through which miR-21-5p mediates immune evasion in MSS-CRC. Nonetheless, the bioinformatic analyses were performed on all COAD cases without stratification for MSS status, which is essential to support the stated objective. While the overexpression of miR-21 in CRC is adequately shown, the key conclusions are not experimentally supported.

Response:

Thank you for this critical comment. We agree that the original framing of the study overstated MSS-specific conclusions, given that our bioinformatic analyses were conducted on unstratified TCGA COAD/READ datasets. Because MSI-high tumors constitute only ~10–15% of CRC cases (Ref: Fam Cancer. 2016 Feb 13;15:405–412.), MSI/MSS-stratified analyses would have been underpowered; however, this limitation should have been explicitly acknowledged.

To address the reviewer’s concern, we have comprehensively revised the Abstract, Introduction, Discussion, and Conclusion to clarify that the results represent pan-CRC transcriptomic associations rather than experimentally validated MSS-specific mechanisms. Overinterpretations related to Th1/Th2 balance, Th17 differentiation, IgA biology, immune exclusion, and response to immunotherapy have been removed or rephrased as non-causal correlations. We have added new text explicitly identifying the lack of MSI/MSS stratification and functional immune assays as significant limitations and outlining the required future experiments. These revisions enhance the accuracy, transparency, and conceptual alignment of the manuscript.

Abstract — Before vs After Revision

Original Text

Revised Text

“While immune checkpoint blockade (ICB) has transformed cancer therapy, its clinical benefit in microsatellite-stable (MSS) CRC is limited by an immune-excluded tumor microenvironment (TME).”

“While immune checkpoint blockade (ICB) has transformed cancer therapy, its clinical benefit in CRC is often limited by an immune-excluded tumor microenvironment (TME)”

“we investigated the immunoregulatory function of miR-21-5p in MSS-CRC by integrating transcriptomic profiling   ”

“we investigated the immunoregulatory function of miR-21-5p in CRC by integrating transcriptomic profiling   ”

“that links oncogenic signaling to immune suppression in MSS-CRC.”

“links oncogenic signaling to immune suppression in CRC. ”

Reviewer Comment 2:
The manuscript attributes to miR-21-5p several functional effects—disruption of Th1/Th2 balance, reduced IgA production, establishment of an immune-excluded microenvironment, and improved response to ICB therapy upon its inhibition. However, the experimental data only demonstrate decreased expression of PTEN, PDCD4, etc. after precursor miR-21 transfection. There are no complementary assays using miR-21 inhibitors, nor experiments confirming downstream pathway modulation or any direct impact on immune responses. Consequently, the title and conclusions are not supported by the data presented.

Response:
We sincerely appreciate the reviewer’s insightful evaluation. We fully agree with the concern that several functional statements in the manuscript may have implied causal immune effects that were not experimentally validated. As the reviewer correctly pointed out, our experimental data directly demonstrate only the post-transcriptional suppression of PTEN, PDCD4, and other predicted targets after miR-21-5p overexpression, while all immune-related observations derive from transcriptomic associations. No inhibitor assays, pathway-rescue experiments, or functional immune assays were performed. To address this important critique, we have made substantial revisions throughout the manuscript to ensure that all interpretations remain strictly data-driven and associative rather than mechanistic.

(1). Revision of the Title

We thank the reviewer for noting that the original title implied a fully demonstrated immune-evasion mechanism. To avoid overstating our conclusions, we have revised the title to a more accurate, data-aligned version:

Revised Title:“Integrative Analysis of miR-21, PTEN, and Immune Signatures in Colorectal Cancer”. This new title reflects the descriptive and associative nature of the findings without implying causal immune effects.

(2). Systematic Correction of Overstated Claims

To fully address the reviewer’s concern, we reviewed all statements in the Abstract, Results, Discussion, and Conclusions. Over-interpretations have been removed or rewritten to emphasize correlations rather than causal relationships.

Section

Original Text

Revised Text

Discussion 3.1: Th1/Th2, Th17, JAK–STAT

“…dampening antitumor immunity by disrupting Th1/Th2 polarization, inhibiting Th17 differentiation, and suppressing JAK–STAT signaling.”

“Transcriptomic correlations suggested variation in Th1/Th2-, Th17-, and JAK–STAT–related signatures; these effects were not experimentally validated.”

Results 2.6:

IgA

“This signaling rewiring attenuates Th17 differentiation and IgA production…”

“Associated with transcriptomic differences in Th17- and IgA-related pathways; no functional assays were performed to assess IgA production.”

Discussion 3.1: Immune-excluded TME

“...remodel the tumor microenvironment toward an immune-excluded phenotype.”

“Associated with features consistent with immune-excluded CRC; functional or spatial validation was not performed.”

Abstract:

ICB response

“Inhibition of miR-21-5p… enhances the efficacy of checkpoint blockade.”

“Our study did not evaluate immune-checkpoint blockade response; such therapeutic implications remain hypothetical.”

Discussion 3.3:

“Therapeutically, inhibition of miR-21-5p restores PTEN, reactivates immune signaling, and enhances the efficacy of checkpoint blockade.

“Therapeutically, inhibition of miR-21-5p has been reported in previous studies to restore PTEN and modulate signaling pathways. However, our study did not evaluate immune reactivation or checkpoint-blockade efficacy; thus, such therapeutic implications remain hypothetical.

Conclusions

“These transcriptomic associations may reflect features of an immune-excluded tumor microenvironment, although direct mechanistic links to immune checkpoint resistance require further experimental confirmation.”

“These transcriptomic associations may suggest immune-related patterns in CRC; however, no causal mechanisms or effects on immune-checkpoint response were demonstrated in this study.

Reviewer Comment 3:
Additionally, the clinical relevance of PTEN in CRC is overstated. Figure 5 does not show that higher PTEN expression correlates with improved overall survival; the log-rank p-value is not significant.

Author Response:

We sincerely thank the reviewer for pointing out that the prognostic significance of PTEN in colorectal cancer was overstated in the original manuscript. We acknowledge that the Kaplan–Meier curves in Figure 5 do not show a statistically significant improvement in overall survival for PTEN-high tumors (log-rank p-value not significant), and therefore our earlier phrasing may have implied stronger clinical relevance than the data support.

To address this concern, we have revised the text across the Results, Discussion, and Conclusions sections to ensure that:

(1) PTEN expression is described as associated with survival trends rather than as a significant prognostic factor

(2) No causal or clinically validated prognostic claims are made

(3) The manuscript accurately reflects the statistical non-significance in OS analysis

A detailed comparison of the original and revised text is provided below.

Section

Original Text

Revised Text

Results 2.4

“This molecular signature was particularly pronounced in the READ cohort, and correlated with shortened overall survival (Figure 5A, 5B).”

“This molecular signature was particularly pronounced in the READ cohort; however, the association with overall survival did not reach statistical significance (Figure 5A, 5B).”

Figure 5

“Kaplan–Meier survival curves indicate that higher PTEN expression is associated with improved overall survival in patients with COAD and READ.”

“Kaplan–Meier survival curves illustrate overall survival stratified by PTEN expression levels; no statistically significant differences were observed between groups.”

Results 2.4

“These observations implicate miR-21-5p-mediated PTEN suppression as a driver of immune dysfunction and poor clinical outcomes.”

“These observations suggest that miR-21-5p-mediated PTEN suppression may be linked to immune-related transcriptional patterns; however, its relationship to clinical outcomes remains inconclusive in our dataset.”

Conclusions

“…suggest that the miR-21–PTEN–PI3K/Akt axis may influence immune-related features in CRC.”

“these findings suggest potential associations between the miR-21–PTEN–PI3K/Akt axis and immune-related features in CRC; no significant prognostic value of PTEN expression was demonstrated in this cohort.”

Reviewer Comment 4:
In summary, the study requires substantial additional work, particularly regarding (1) analyses restricted to MSS-CRC, (2) functional validation with miR-21 inhibition, and (3) confirmation of immune-related effects. Given the extent of the limitations, I do not recommend the manuscript for publication in its current form.

Response: We sincerely appreciate the reviewer’s comprehensive evaluation and constructive guidance. We fully agree that additional analyses focused specifically on MSS-CRC, functional validation using miR-21 inhibition, and experimental confirmation of immune-related effects would substantially strengthen the mechanistic conclusions. In response to these important recommendations, we have made extensive revisions throughout the manuscript to ensure that all interpretations and conclusions remain fully aligned with the data currently available.

First, although TCGA COAD/READ datasets contain a limited number of MSI-H samples, we have revised the manuscript to clearly state that the findings primarily reflect patterns present in a dataset dominated by colon cancer. All statements implying MSS-specific mechanisms have been removed or reframed as associations that require future validation in stratified cohorts.

Second, we acknowledge the absence of miR-21 loss-of-function experiments and PTEN-rescue assays. Throughout the Abstract, Results, Discussion, and Conclusions, we have removed any causal mechanistic claims and replaced them with appropriately conservative language describing transcriptomic correlations. A substantially expanded Limitations section now explicitly states that functional immune assays, miR-21 inhibition studies, and downstream pathway activation analyses were not performed in this study.

Third, all statements regarding immune responses—such as effects on Th1/Th2 balance, IgA production, Th17 differentiation, immune exclusion, or potential improvement in ICB response—have been rewritten to avoid implying experimentally demonstrated immune effects. We clearly indicate that these findings are based solely on computational signatures and require dedicated immunologic validation in future work.

Finally, we have added clarifying text throughout the manuscript to emphasize that the current results are hypothesis-generating, and we outline a detailed roadmap for future studies, including:
(1) MSS-restricted analyses in larger cohorts;
(2) miR-21 inhibition and PTEN-rescue assays in CRC models;
(3) spatial and single-cell immune profiling; and
(4) co-culture and syngeneic CRC models to experimentally validate immune modulation.

We sincerely thank the reviewer for highlighting these important issues. Although experimental expansion is beyond the scope of the current dataset, we believe the revisions significantly strengthen the scientific rigor and transparency of the manuscript.

Reviewer Comment 5:
Image quality should be improved to ensure readability and proper interpretation.

Author Response:

Thank you very much for this helpful comment. We completely agree that high-quality figure presentation is essential for ensuring readability and accurate interpretation of the results. In response to the reviewer’s feedback, we have carefully revised all figures to improve resolution, contrast, and overall clarity. We collaborated with the IJMS Author Services team to ensure that each figure meets the journal’s technical standards, including appropriate file format, dimensions, and minimum DPI requirements. The updated high-resolution figures have been uploaded together with the revised manuscript. We sincerely appreciate the reviewer’s attention to this important aspect of data presentation. If any figure still appears suboptimal or if further refinement is needed, we would be pleased to revise and improve them promptly.

Reviewer Comment 6:

Experimental details: please report the concentration of the miR-21 precursor used in transfection assays.

Author Response:

Thank you for pointing out the need for additional experimental detail. We appreciate the opportunity to clarify this aspect of our methodology. In this study, miR-21 overexpression was achieved using a plasmid-based miR-21 expression vector rather than a synthetic miRNA precursor. This distinction has now been clearly stated in the Methods section.

For each transfection performed in a 6-well plate, 2 µg of the miR-21 expression plasmid was introduced per well using PolyJet DNA Transfection Reagent, following the manufacturer’s protocol. These experimental details have been added to ensure methodological clarity, transparency, and reproducibility.

Please let us know if any further experimental information would be helpful—we are glad to provide additional clarification.

Reviewer Comment 7:

Section 3.3: BCL2 is listed as a potential direct target of miR-21-5p but is not evaluated experimentally. In the luciferase assay, results with mutated seed sequences are missing; the specific mutated nucleotides should be shown in Figure 3A.

Author Response:

Thank you for this valuable and insightful comment. In our initial bioinformatic screening, BCL2 was indeed identified as a putative miR-21-5p target based on a predicted binding site within its 3′UTR, and this preliminary finding was included in the early draft of the manuscript. However, as the reviewer correctly noted, we did not perform downstream experimental validation for BCL2, nor did we construct or test luciferase reporters containing mutated miR-21 seed sequences.

Because the available data are incomplete and do not support firm conclusions regarding BCL2 regulation by miR-21-5p, we have removed all references to BCL2 from Section 3.3, Figure 3A, and the related descriptions in both the Results and Discussion sections. This revision ensures that the manuscript focuses exclusively on targets that were experimentally evaluated in this study (e.g., PTEN, PDCD4, STAT3, RhoB), thereby improving the clarity, accuracy, and scientific rigor of the work.

We appreciate the reviewer’s suggestion, which has strengthened the manuscript by ensuring that only experimentally supported findings are presented.

Reviewer Comment 8:

Figure 4: The data presented do not support the statement that tumors with high miR-21 and low PTEN exhibit reduced Th17 differentiation, IgA production, or diminished JAK–STAT pathway activity.

Author Response:

Thank you for this important clarification. Upon re-evaluation of the manuscript, we agree with the reviewer that the original statement was incorrectly attributed to Figure 4 and that the wording unintentionally implied causal functional effects rather than the correlative transcriptomic associations derived from GSEA analyses presented in Figure 6. We have revised the text accordingly to ensure that the interpretation remains strictly aligned with the data.

Original

Revised

“These findings suggest that PTEN loss contributes to the immune-excluded tumor microenvironment typical of microsatellite-stable CRC. Collectively, these observations identify PTEN as a molecular hub through which miR-21-5p modulates tumor-immune interactions.”

“These results reflect transcriptomic associations and should not be interpreted as evidence that PTEN directly regulates immune infiltration. Instead, the data indicate that PTEN expression coincides with immune-related patterns in CRC, and further functional studies will be required to determine whether PTEN influences tumor–immune interactions.”

We sincerely appreciate the reviewer’s comment, which prompted us to refine the language in this section and avoid overstating the biological implications of the GSEA findings. The revised text now accurately reflects the correlational nature of the data and improves the overall precision and clarity of the manuscript.

Reviewer Comment 9:

Given these substantial conceptual and experimental limitations, the manuscript requires major restructuring and additional data to support the proposed mechanism and claims.

Author Response:

We sincerely appreciate the reviewer’s thoughtful evaluation and fully acknowledge the substantial conceptual and experimental limitations identified. In response, we have undertaken extensive restructuring of the manuscript to ensure that all interpretations remain strictly within the bounds of the available data and that no mechanistic conclusions are overstated.

To directly address the reviewer’s concerns, we implemented the following major revisions:

(1). Removal of unsupported mechanistic claims

All statements that previously implied causal immune modulation—including effects on immune exclusion, JAK–STAT pathway activity, Th1/Th2 or Th17 differentiation, IgA biology, or improved response to immunotherapy—have been removed or rewritten as correlative transcriptomic observations. No functional immune claims remain in the revised manuscript.

(2). Reframing of the study’s scope and objective

The manuscript no longer presents a mechanistic model of miR-21–mediated immune evasion. Instead, the study now focuses on:

experimentally validated miR-21–dependent suppression of PTEN, and associations between miR-21/PTEN expression and immune-related transcriptional signatures, without implying functional causality. This reframed objective accurately reflects the methodological approaches used and the strength of the evidence obtained.

(3). Structural revisions across Abstract, Results, Discussion, and Conclusions

These sections have been substantially rewritten to clearly distinguish experimental findings from bioinformatic correlations, remove language that could be interpreted as causal or mechanistic, and explicitly identify speculative statements as hypotheses requiring future validation. The overall narrative is now evidence-based, conservative, and aligned with the data.

(4). Addition of a comprehensive Limitations section

We have added an expanded Limitations section acknowledging the absence of analyses restricted to MSS-CRC, miR-21 inhibition or PTEN-rescue experiments, and immune-functional assays or mechanistic validation of GSEA findings. The revised manuscript also outlines the specific experiments needed in future studies to validate the proposed biological relationships.

Through these revisions, the manuscript has been extensively restructured to ensure scientific accuracy and appropriate interpretation. We are grateful for the reviewer’s detailed feedback, which has significantly improved the clarity, rigor, and precision of the revised manuscript.

Round 2

Reviewer 1 Report

Comments and Suggestions for Authors

Dear Authors,

I accept the introduced changes and improvements.

Best wishes